# Tolerability of Atovaquone—Proguanil Application in Common Buzzard Nestlings

**DOI:** 10.3390/vetsci9080397

**Published:** 2022-07-30

**Authors:** Anja Wiegmann, Tony Rinaud, Meinolf Ottensmann, Oliver Krüger, Andrea Springer, Marko Legler, Michael Fehr, Christina Strube, Nayden Chakarov

**Affiliations:** 1Department of Animal Behaviour, Bielefeld University, Konsequenz 45, 33615 Bielefeld, Germany; anja.wiegmann@tiho-hannover.de (A.W.); tony.rinaud@uni-bielefeld.de (T.R.); m.ottensmann@uni-bielefeld.de (M.O.); oliver.krueger@uni-bielefeld.de (O.K.); 2Institute for Parasitology, Centre for Infection Medicine, University of Veterinary Medicine Hannover, Buenteweg 17, 30559 Hannover, Germany; andrea.springer@tiho-hannover.de (A.S.); christina.strube@tiho-hannover.de (C.S.); 3Department of Small Mammal, Reptile and Avian Diseases, University of Veterinary Medicine Hannover, Buenteweg 9, 30559 Hannover, Germany; marko.legler@tiho-hannover.de (M.L.); michael.fehr.ir@tiho-hannover.de (M.F.)

**Keywords:** Malarone, drug tolerance, avian malaria, haemosporidian parasites, blood chemistry, *Buteo buteo*

## Abstract

**Simple Summary:**

Many wild animals, and particularly birds, are commonly infected and can suffer health consequences by blood parasites related to *Plasmodium*, the causative agents of malaria in humans. Atovaquone–proguanil (Malarone^®^, GlaxoSmithKline) is one of the most popular drugs for the treatment of malaria infections in humans and is commonly used for the treatment of birds in captivity. Our aim was to test the potential effects of Malarone^®^ within one week of treatment on the growth rate, body condition, and blood chemistry of common buzzard nestlings, a widely distributed Eurasian bird of prey. We found no evidence of detrimental effects of a single dose in common buzzard nestlings with an average dosage of 11 mg/kg, compared with the 7 mg/kg recommended daily dosage in humans. Although Malarone^®^ is commonly used in wildlife rehabilitation centres, and our results do not indicate acute toxicity, further studies are needed to determine the half-life and potential long-term effects of Malarone^®^ treatment in birds.

**Abstract:**

Differences in drug tolerability among vertebrate groups and species can create substantial challenges for wildlife and ex situ conservation programmes. Knowledge of tolerance in the use of new drugs is, therefore, important to avoid severe toxicity in species, which are both commonly admitted in veterinary clinics and are of conservation concern. Antimalarial drugs have been developed for use in human medicine, but treatment with different agents has also long been used in avian medicine, as haemosporidian infections play a major role in many avian species. This study investigates the effects of the application of atovaquone–proguanil (Malarone^®^, GlaxoSmithKline) in common buzzards (*Buteo buteo*). The potential effects of treatment on body condition, growth rate, and chemical blood parameters of nestlings were assessed. All individuals survived the treatment, and no effects on body condition, growth rate, and chemical blood parameters were observed. Our results suggest the tolerability of Malarone^®^ in common buzzards at a single dose of on average 11 mg/kg body weight. For its safe use, we recommend further studies to determine pharmacokinetics in different avian species as well as to assess the effects of repeated treatment.

## 1. Introduction

Starting in the middle of the 20th century, avian malaria was used as a model for human malaria research for about 20 years, including studies of various antimalarial drugs [1]. In addition to the continuing experimental studies in birds, these drugs are also used for the treatment of haemosporidian infections in poultry and birds in zoos and rehabilitation centres [2]. Due to the possibility of a severe disease course, vector-borne haemoparasites, such as the causative agents of malaria, *Plasmodium* spp., and other closely related genera, can play major roles in the health development of captive birds [3]. Encounters between non-coevolved hosts and parasites are, for example, caused by captivity and induced by humans. Exposure to vectors and blood parasites without the possibility of using natural parasite-avoidance strategies (e.g., migration, feeding, roosting, and nest site choice [4]) can make context-dependent antiparasitic treatments necessary [5]. To treat *Plasmodium* infections in avian species, the application of chloroquine and primaquine or combination therapies are routinely proposed, although reports about liver toxicity and even death as a result of this treatment in birds exist [6,7]. Besides selecting for resistance, these substances can have substantial side effects in humans as well as in other mammals and birds [8,9,10,11], invoking the need for taxon-specific analyses of drug safety [12]. The synergistic combination of atovaquone and proguanil (Malarone^®^, GlaxoSmithKline GmbH & Co. KG, München, Germany) has shown efficacy against hepatic and erythrocytic stages of *Plasmodium falciparum* with only few and mild adverse effects (e.g., vomiting, diarrhoea, and increases in hepatic transaminases) in adults and children [13]. Both components affect two different parasitic pathways: Atovaquone inhibits mitochondrial electron transport, whereas cycloguanil, as the primary metabolite of proguanil, inhibits the parasite’s dihydrofolate reductase [14]. Studies and case reports on the use of Malarone^®^ in birds are still rare and mostly focused on a reduction in parasite burden [15,16,17] without assessing the potential consequences of treatment on the physiology of the vertebrate host. Based on an extensive long-term study of a common buzzard (*Buteo buteo*) population in eastern Westphalia, Germany, the present study addresses the tolerability of Malarone^®^ in this common European raptor species, by investigating the effects of treatment on the health of juvenile wild raptors via an evaluation of body condition, growth rate, and blood chemistry.

## 2. Materials and Methods

### 2.1. Study Population

Data included in this study were collected as part of a larger experiment investigating host–parasite interactions between common buzzards and their blood parasites, conducted between 2016 and 2020 in an intensively monitored population in eastern Westphalia, Germany (see [18] and [19] for a detailed description of the study system). Sampling and drug application were permitted by the ethics committee of the Animal Care and Use Committee of the German North Rhine-Westphalia State Office for Nature, Environment, and Consumer Protection (*Landesamt für Natur, Umwelt und Verbraucherschutz Nordrhein-Westfalen*) under reference numbers 84-02.04.2017.A147 and 84-02.04.2014.A091.

### 2.2. Malarone^®^ Treatment

A total of 299 nestlings were sampled twice in 2016 (*n* = 39), 2018 (*n* = 70), 2019 (*n* = 73), and 2020 (*n* = 117), with a median interval of 7 days (SD = 4.16, 4–25) between the sampling sessions. Nestlings were randomly assigned to the following treatment groups: (i) control (standard sampling protocol, without any treatment, *n* = 150); (ii) water treatment (1 mL of tap water applied orally, *n* = 53); and (iii) Malarone^®^ treatment (see below for details; *n* = 96). Based on the recommended dose of 7 mg/kg body weight [15,17], each nestling belonging to the treatment group was treated orally with a syringe into the throat with a fixed dose of 7 mg Malarone^®^ (corresponding to approx. 5 mg atovaquone and 2 mg proguanil hydrochloride), dissolved in 1 mL of tap water. This fixed dose was used in every nestling due to faster handling and, thus, stress reduction in the wild birds. Nestlings were, on average, observed for 5 min after treatment before the birds’ return to the nest.

### 2.3. Body Condition and Growth Rate

For all individuals, body weight (to the nearest 5 g) and wing length (to the nearest mm) were recorded before treatment and during the second sampling session. Wing length was used to estimate nestling age based on sex-specific standard growth curves [20]. Average daily growth rates were calculated based on the change in body weight between the first and second sampling, while accounting for the variable time interval between the two measurements, using the following formula: growth rate = change in body weight/days between sampling sessions. Body condition indices were estimated as the residuals of a sex-specific linear regression of weight on wing length based on standard growth data [20]. As groups differed in the mean body condition prior to the treatment (i.e., at the first sampling), we computed the change in body condition by subtracting the values of the first sampling from the values at resampling. To improve model convergence and the interpretation of model estimates, both variables were standardised using a z-transformation [21].

### 2.4. Blood Chemistry

All individuals examined for blood chemistry values were part of a previous study that analysed changes in blood chemistry in response to *Leucocytozoon* infection [22]. Here, we expanded the dataset by comparing blood chemistry values of Malarone^®^-treated individuals before and after resampling. In May and June 2019, 11 nestlings were sampled in order to examine the tolerance towards Malarone^®^ (GlaxoSmithKline GmbH & Co., KG, München, Germany) for a suite of blood chemistry values. A sample of 0.5 mL blood was collected from the ulnar vein in 1.3 mL heparinised tubes (Sarstedt AG & Co., KG, Nümbrecht, Germany) and centrifuged within 30 min after sampling. Plasma was separated from blood cells and stored at −80 °C until analyses. Treated nestlings were revisited after 7 days for a post-treatment blood sample. Blood chemistry analyses were conducted with SYNLAB.vet using an AU680 Clinical Chemistry Analyser (Beckman Coulter). The following blood chemistry parameters were measured: alkaline phosphatase (AP), aspartate aminotransferase (AST), lactate dehydrogenase (LDH), gamma-glutamyl transferase (GGT), creatine kinase (CK), butyrylcholinesterase (BuChE), bile acids (BA), albumin (ALB) and total protein (TP). In the absence of a placebo-treated control group, we resorted to another 49 nestlings sampled a single time, originally as part of another study (see above), to obtain baseline blood chemistry values of individuals in a comparable age range to those prior to Malarone^®^ treatment (*n* = 38, referred to as “Pre-Control”) and 7 days post-treatment (*n* = 11, “Post-Control”). 

### 2.5. Statistical analyses

All statistical analyses were conducted in R version 4.1.0 [23]. To analyse the treatment effects on two fitness proxies (the change in body condition and growth rate), two linear mixed models were implemented using the lme4 package [24]. All models met the equivariance and independence of the residuals as well as residuals’ normality. Models were implemented using the change in body condition and growth rate as the response variables. As fixed effects, we specified the treatment groups, year of sampling (to estimate inter-annual variation), sex (only for growth rate), and average nestling age (i.e., age at the midpoint of the sampling interval). Additionally, we accounted for variable sampling intervals by using the days between samplings as a covariate. Nest ID was fitted as a random factor in all models to consider the non-independence of siblings. Furthermore, a non-parametric Mann–Whitney test was used to investigate mean rank differences in nine different blood chemistry parameters. Therefore, we compared both pre-treatment (“Pre-Control” vs. “Pre-Mal”) and post-treatment (“Post-Control” vs. “Post-Mal”) groups, respectively. To account for multiple testing, we computed adjusted *p*-values using Benjamini–Hochberg correction. Additionally, we performed a non-parametric Wilcoxon’s signed-rank test to identify potential systematic changes in blood chemistry values between pre- and post-treatment for the subset of Malarone^®^-treated individuals. We corrected for multiple testing as before.

## 3. Results

Regurgitation of Malarone^®^ or water treatment was not observed within the 5 min of observation after application. Since a fixed quantity of 7 mg was administered to every chick, a mean dose of 11 mg/kg was ultimately administered (lowest dose 7 mg/kg, highest dose 21 mg/kg). The youngest nestling treated was 12 days old, and the oldest 29 days (mean 19 days), while their weight ranged between 335 g and 990 g (mean 639 g).

All individuals allocated to the Malarone^®^ treatment survived until resampling. There was no evidence for a difference in the development of body condition among treatments (control vs. water, est. = −0.036, *p* = 0.84; control vs. Malarone^®^: est. = 0.047, *p* = 0.77) after accounting for age and among-year variation (Table 1, Figure 1, Appendix A). The growth rate did not differ among treatment groups, after controlling for age, sex, and year of sampling (Table 1, Figure 1, Appendix A). Further, the growth rate appeared to be age- and sex-dependent, whereby females and younger chicks showed faster growth rates (Figure 2). 

The examination of nine blood chemistry values did not reveal any differences in pairwise comparisons of pre- and post-treatment groups (Mann–Whitney tests, all unadjusted *p*-values > 0.081, all adjusted *p*-values > 0.390, Appendix A, Figure 3, Appendix A). Changes in blood chemistry values between Malarone^®^ pre- and post-treatment groups were apparent prior to *p*-value correction for AP, GGT, and LDH. However, no significant differences were identified when using Benjamini–Hochberg correction (Appendix A).

## 4. Discussion

In avian medicine, considerable differences in drug tolerance demonstrate the necessity of species-specific assessments for the safe use of drugs [25]. Several examples of serious side effects due to medication exist. For instance, the application of the antimycotic drug voriconazole leads to clinical signs and toxicity in different penguin species [26]. Similarly, medication with itraconazole results in serious toxicity in African grey parrots (*Psittacus erithacus*) [27], and the antibiotic gentamycin shows differences in pharmacokinetics among bird species [28] as well as adverse effects such as an increase in chemical blood parameters [29]. Probably, one of the most popular examples of drug toxicity in raptors is the intake of the non-steroidal anti-inflammatory drug (NSAID) diclofenac and its lethal effect on different vultures [30]. In these raptors, a single exposure already leads to death within 48 h [31]. 

In the present study, no negative effects of treatment with Malarone^®^ were found on the body condition, growth rate, or blood chemistry values of common buzzards one week after treatment. The detected age- and sex-dependent growth rate is in line with previous findings, as growth follows a logarithmic function, according to which females reach a larger size than males and younger chicks grow quicker [20]. Moreover, variations in growth rate and body condition could not be attributed to medication but were affected by sampling year, as well as by differences among broods. Comparable results in related studies on adult greenfinches (*Carduelis chloris*) [17] and blue tits (*Cyanistes caeruleus*) [15] also did not show any differences in body condition between Malarone^®^-treated and control animals. Experimental trials by Knowles et al. [15] linked Malarone^®^ application in breeding adults with nest desertion, but similar effects were not expected or observed in the present study, where only nestlings were treated. As our study was performed on tree-nesting and sensitive wild birds, no monitoring of possible further side effects could be performed during the one-week treatment period or thereafter, and thus, such effects cannot be excluded. However, since liquids are quickly transferred into the intestines of birds [32], the fast absorption of Malarone^®^ can be assumed. However, the half-life of Malarone^®^ in common buzzards and especially in their nestlings is unknown. Compared with humans, the elimination half-life of atovaquone in paediatric patients of 1–2 days is lower than that in adults with 2–3 days, while the elimination half-life of proguanil is equal (12–21 h) [33]. As we cannot accurately predict whether a 7-day control interval is long enough to examine all potential side effects, detailed follow-up studies, taking the correlation to the real elimination time into account, are necessary. 

To our knowledge, no pharmacokinetic studies have been performed regarding the atovaquone–proguanil combination in raptors, so the dosing was based on recommendations for humans and other avian species and was finally adopted as a fixed dose for a rapid application to minimise the stress level of the wild birds. With a single application of 11 mg/kg on average, we did not detect negative impacts on the growth rate and body condition of the raptor nestlings. We acknowledge that using a fixed dose for every nestling is a limitation of the current study, as possible dose-dependent negative effects might be absent in larger chicks, which, in turn, could potentially mask negative effects in smaller chicks. This was dictated by the primary aim of the treatment of securely suppressing haemosporidian infections with a single dose but included the risk of overdosing. Future studies with a better possibility of experimental group control may prove to be additionally informative through the use of weight-adjusted dosages in several dosage groups of age-controlled raptors or other avian subjects. In other species, including humans, Malarone^®^ is administered over several days, accounting for the short half-life of this drug, and the dose varies fourfold between prophylaxis and treatment [33]. The highest administered dose of this study was 21 mg/kg lower than treatment dosage recommendations for children, calculated down from the lowest existing dose recommendation of 5 kg body weight [33]. However, this can still only be considered as a rough comparison due to the presumably existing differences in the drug clearance between common buzzards and humans. Previous studies on passerine birds used significantly longer treatment periods, without serious side effects on the treated individuals [15,17]. Nonetheless, to avoid side effects and toxicity, it is usually advantageous to determine the lowest effective dose of drugs [34]. Furthermore, depending on the aim of the treatment, single applications may be preferable, since repeated access to wild birds for multiple dosing may be difficult, and wild birds may be particularly stressed by human contact [35]. Additionally, a recent study on the same buzzard population by Rinaud et al. [36] demonstrated a successful reduction in *Leucocytozoon* infection intensity with a single application of Malarone^®^, despite its incomplete elimination from the birds’ blood.

Relevant differences in pharmacokinetics and pharmacodynamics exist not only between mammals and birds with great differences in anatomy and physiology [12,37,38] but also between bird taxa. For example, great differences in the elimination rate of three different NSAIDs were shown among chickens (*Gallus gallus*), turkeys (*Meleagris gallopavo*), mallards (*Anas platyrhynchos*), ostriches (*Struthio camelus*), and feral pigeons (*Columba livia domestica*) [39]. In a recent study, comparable pharmacokinetics between humans and birds were shown for primaquine [40], the antimalaria drug commonly used in avian medicine [6]. Comparable studies are still lacking for atovaquone and proguanil. Atovaquone, the main ingredient of Malarone^®^, is highly protein-bound after its absorption and is eliminated almost exclusively via the liver, without any known metabolism [41]. Proguanil is partly metabolised to cycloguanil via isoenzyme 2C19 of cytochrome P450 (CYP) [41]. Some studies in avian medicine suggest evolutionary variation in CYP, which is responsible for the xenobiotic metabolism of synthetic drugs, resulting in species-specific resistance as well as sensitivity to various agents [42,43]. Additionally, variations in plasma protein-binding capacity between different species may cause different plasma concentrations of drugs, as in the case of the highly plasma-protein-bound antifungal drug itraconazole [44]. 

Although caution is still warranted due to the small number of individuals with examined blood chemistry in the present study, we interpret our results as the first indication of the tolerability of Malarone^®^ in common buzzards, one of the most common raptor species receiving veterinary attention in Europe. Compared with human medicine, elevated levels of alanine aminotransferase (ALT) and aspartate aminotransferase (AST) are among the possible adverse side effects of Malarone^®^ treatment [33], but these are known as reversible and not associated with clinical signs [45]. However, when evaluating changes in blood chemistry, it should also be considered that existing haemosporidian infections may account for additional variation [22,46,47].

In conclusion, this study gives the first indication that Malarone^®^ application is safe for clinical or experimental use against haemosporidian infections in common buzzards, even in two-week-old nestlings. We recommend follow-up studies on the tolerability of Malarone^®^ in different avian species, multiple dosing, as well as on its efficacy against different haemosporidian groups.

## Figures and Tables

**Figure 1 vetsci-09-00397-f001:**
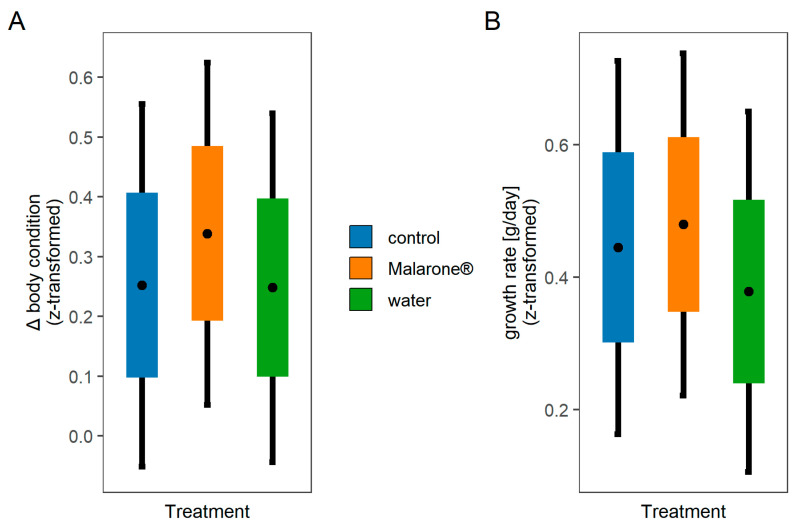
Effects of treatment and control on body condition and growth rate. Visualisation of the predicted values of (**A**) body condition and (**B**) growth rate from linear mixed models testing the effect of treatment (control, Malarone^®^, and water) on these two fitness proxies. Black dots represent mean estimates for each of the treatment groups. Main boxplots represent standard deviations, and error bars indicate 95% confidence intervals. See Table 1 for statistical comparison of treatment groups.

**Figure 2 vetsci-09-00397-f002:**
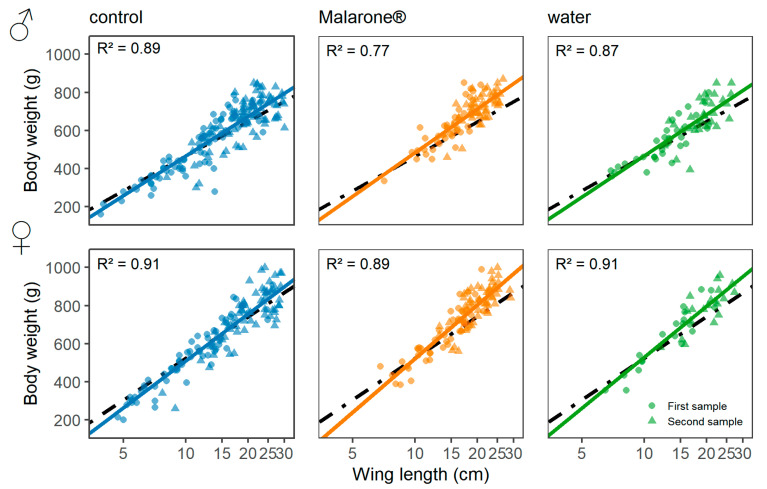
Regression analyses for weight and wing length. Visualisation of regression analyses for weight and wing length of control, Malarone^®^-, and water-treated nestlings (solid line) in comparison to the reference data of common buzzards [20] (dotted black line) for female (󠇂♀, *n* = 136) and male (♂, *n* = 163) individuals. Circles represent values at first sampling, triangles at second sampling. R^2^ represents the proportion of variance explained by both fixed and random effects in each model.

**Figure 3 vetsci-09-00397-f003:**
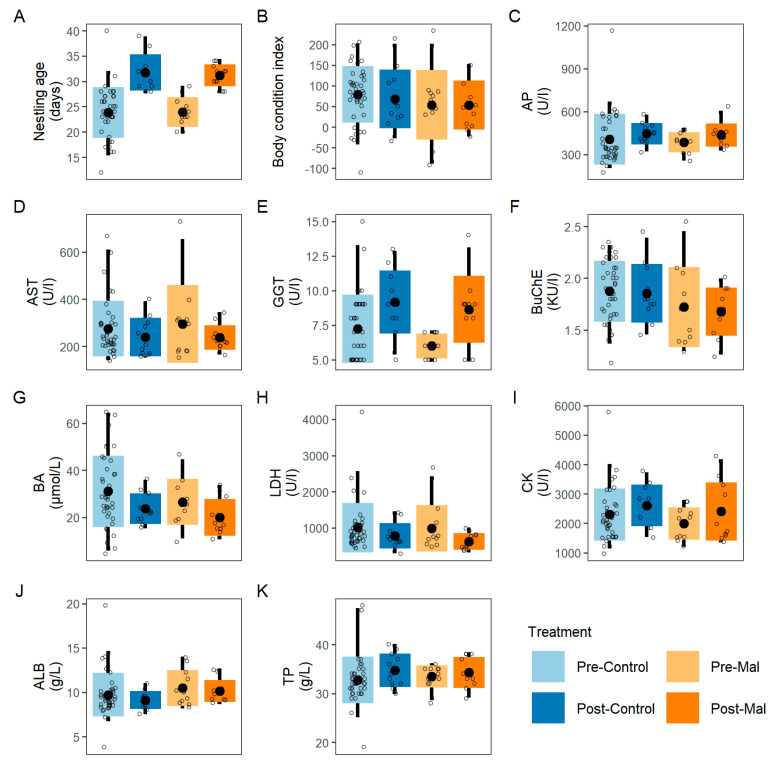
Mean differences of eleven parameters among treatment and control. Visualisation of mean differences in nestling age at sampling (**A**), body condition index (**B**), and blood chemistry parameters (**C**−**K**) among four groups: “Pre−Control”, “Post−Control”, “Pre−Mal”, and “Post−Mal”. Black dots represent mean estimates for each of the treatment groups. Main boxplots represent standard deviations, and error bars indicate 95% confidence intervals. Open circles represent raw data for each individual. See Appendix A for statistical comparisons of parameters (**B**−**K**).

**Table 1 vetsci-09-00397-t001:** Treatment effects on body condition and growth rate of common buzzards.

Predictors	Δ Body Condition				Growth Rate			
	Estimates	CI	*t*-Value	*p*	Estimates	CI	*t*-Value	*p*
Intercept	0.41	−0.53–1.35	0.86	0.391	2.83	2.02–3.64	6.91	**<0.001**
Mean Age (days) *	−0.01	−0.04–0.02	−0.42	0.676	−0.09	−0.12–−0.07	−6.91	**<0.001**
Resampling interval (days)	0.00	−0.07–0.08	0.13	0.895	−0.02	−0.08–0.04	−0.68	0.498
Sex (Males)					−0.40	−0.61–−0.18	−3.67	**<0.001**
Treatment (Control)	*Reference*				*Reference*			
Treatment (Water)	−0.04	−0.39–0.32	−0.20	0.841	−0.09	−0.41–0.23	−0.55	0.580
Treatment (Malarone^®^)	0.05	−0.27–0.37	0.29	0.774	0.01	−0.28–0.29	0.04	0.970
Year (2020)	*Reference*				*Reference*			
Year (2016)	−1.46	−3.50–0.58	−1.41	0.159	−1.02	−2.81–0.76	−1.13	0.260
Year (2018)	−0.38	−0.75–−0.02	−2.08	**0.039**	−0.28	−0.59–0.03	−1.78	0.077
Year (2019)	−0.56	−0.95–−0.17	−2.83	**0.005**	−0.44	−0.76–−0.11	−2.62	**0.010**
**Random Effect**								
σ^2^	0.65				0.62			
ICC	0.38				0.23			
**N**	109 Broods				109 Broods			
**Observations**	261				261			
**Marginal R^2^/Conditional R^2^**	0.058/0.416				0.267/0.439			

Linear mixed model results testing for treatment effects on body condition (change between sampling events, z-transformed) and growth rate (z-transformed) of common buzzard nestlings. Nestling age and year of sampling are additional fixed effects as well as nestling sex for the growth rate model. Significant *p*-values are printed in bold. * Midpoint of the sampling interval.

## Data Availability

R code and documentation are available as a PDF file written in RMarkdown (Appendix A) and available along with the raw data on GitHub (https://github.com/TonyRinaud/malarone_tolerability, accessed on 31 July 2022).

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
