# Peer review of "Tolerability of Atovaquone—Proguanil Application in Common Buzzard Nestlings"

_vetsci, 2022, doi:10.3390/vetsci9080397_

Round 1
Reviewer 1 Report
This is a relatively straightforward and interesting study, and I commend the authors for their interest in investigating whether malarone is safe for buzzards.
My main concern with the study is that the authors state that they based dosage on a recommended dose of 7 mg/kg, yet they administered 7 mg/bird regardless of the individual bird mass. It seems the nestlings ranged anywhere from 335 g to 900 g (as reported on line 132), therefore the doses received by the smaller chicks would have been up to three times greater (!!!) than the recommended dose, which is especially concerning given that these younger chicks were not at their full physiological capacity to detoxify/eliminate potentially toxic drugs/byproducts. Fortunately based on the study results it seems the birds were okay afterall, but I do wonder if these high doses were indeed covered by the ethics approval obtained by the authors. To make sure that there is adequate compliance with the journal's ethical standards, I would recommend that the authors provide a copy of the ethics submission and approval documentation so it can be verified that the procedures in the study did indeed follow what had been authorized by the ethics committee.
My second concern, assuming that the journal's ethical standards are met, is that since there was a broad variation in dose the statistical analyses need to take this into account and the comparison of proxies for fitness (condition and growth rate) and physiological (plasma chemistries) need to be adjusted relative to dosage. Otherwise it is possible that negative effects that did occur in smaller chicks (which received higher doses) are being statistically masked by the absence of negative effects in larger chicks.
OTHER REMARKS
Line 29 - I think the phrasing "Already since" is not a fair statement... Yes, the 1950-1970s avian malaria was widely used as a model for human malaria research. However, since the 1980s it was replaced by murine malaria models, and at present avian malaria has lost its significance as a model for human malaria.
Line 39 - Reference 6 does not mention resistance to chloroquine/primaquine. Reference 7 only mentions resistance to chloroquine/primaquine in human-infecting Plasmodium. Since the Plasmodium that infect birds and mammals are considerably different, and to my knowledge there is no report of chloroquine/primaquine resistance in bird-infecting Plasmodium, this sentence is not appropriate. There are, however, several instances documenting liver toxicity and even death as a result of the chloroquine/primaquine treatment in birds (mentioned by both references 6 and 7), which is perhaps a better argument to justify seeking other terapeutic options.
Line 242 - The question is not how quickly it is absorbed, but rather how quickly it is inactivated/eliminated. In humans, the half-life of malarone is 2-3 days (Beerahee, M., 1999. Clinical pharmacology of atovaquone and proguanil hydrochloride. Journal of Travel Medicine 6:S13), which means that at 7 days you'd still have a residual dose that could range from 6 to 25% of the original dose. That's assuming that the clearance rate in buzzard nestlings is similar to that of adult humans, which is something we do not know and probably is not true. The validity of this post-control group therefore needs to be discussed more critically considering the pharmacokinetics of malarone (as overviewed in lines 260-276).
Line 247 - As indicated above, this needs to consider a dose-response relationship. You cannot just look at the average dose and claim there was no effect, you actually need to check if there was an effect at higher doses.
Line 248 - "Development" is a broad term, and you did not really verify develpment as a whole. It is possible that there were longer-term developmental effects that were not detected by the study design. Therefore this should be replaced by "growth rate and body condition".
Author Response
Reviewer 1
My main concern with the study is that the authors state that they based dosage on a recommended dose of 7 mg/kg, yet they administered 7 mg/bird regardless of the individual bird mass. It seems the nestlings ranged anywhere from 335 g to 900 g (as reported on line 132), therefore the doses received by the smaller chicks would have been up to three times greater (!!!) than the recommended dose, which is especially concerning given that these younger chicks were not at their full physiological capacity to detoxify/eliminate potentially toxic drugs/byproducts. Fortunately based on the study results it seems the birds were okay afterall, but I do wonder if these high doses were indeed covered by the ethics approval obtained by the authors. To make sure that there is adequate compliance with the journal's ethical standards, I would recommend that the authors provide a copy of the ethics submission and approval documentation so it can be verified that the procedures in the study did indeed follow what had been authorized by the ethics committee.
We thank for the opportunity to make this clearer. Attached you can find the ethics submission and the approval documentation permitted by the ethics commission of the Animal Care and Use Committee of the German North Rhine-Westphalia State Office for Nature, Environment and Consumer Protection (Landesamt für Natur, Umwelt und Verbraucherschutz Nordrhein-Westfalen).
My second concern, assuming that the journal's ethical standards are met, is that since there was a broad variation in dose the statistical analyses need to take this into account and the comparison of proxies for fitness (condition and growth rate) and physiological (plasma chemistries) need to be adjusted relative to dosage. Otherwise, it is possible that negative effects that did occur in smaller chicks (which received higher doses) are being statistically masked by the absence of negative effects in larger chicks.
We are grateful for the suggestion and possibility to consider potential effects of different dosaging. We have now added supplementary figures comparing chicks which received comparatively higher and lower dosage of Malarone. The figures suggest no significant differences and rather higher growth rates in chicks which received a higher dose of Malarone – a pattern which is explained by these chicks being younger and because of this growing relatively faster.
L 29-31: We are grateful for this suggestion revised the information.
L 39-24: We agree and changed the lines as suggested.
L 242: The question is not how quickly it is absorbed, but rather how quickly it is inactivated/eliminated. In humans, the half-life of malarone is 2-3 days (Beerahee, M., 1999. Clinical pharmacology of atovaquone and proguanil hydrochloride. Journal of Travel Medicine 6:S13), which means that at 7 days you'd still have a residual dose that could range from 6 to 25% of the original dose. That's assuming that the clearance rate in buzzard nestlings is similar to that of adult humans, which is something we do not know and probably is not true. The validity of this post-control group therefore needs to be discussed more critically considering the pharmacokinetics of malarone (as overviewed in lines 260-276).
We are grateful for this comment and the opportunity to improve the discussion. The mentioning of the rapid absorption in birds was intended as support of the low probability of regurgitation of the drug. Regarding the control interval of 7 days, we critically extended the discussion (Ll. 221 -227).
L 247: As indicated above, this needs to consider a dose-response relationship. You cannot just look at the average dose and claim there was no effect, you actually need to check if there was an effect at higher doses.
We revised as suggested from your second concern. Please see our response above.
L 249: Revised as suggested.

Reviewer 2 Report
In the manuscript “Tolerability of atovaquone-proguanil application in common 2 buzzards” the authors carried out a treatment of nestling with Malarone® and evaluated body condition, growth rate and 19 chemical blood parameters. The manuscript is well presented but some minor point should be addressed by the authors.
1. Why did the authors choose buzzards as experimental model? Could the authors elaborate in the introduction section?
2. For a better understanding, I highly suggest the division of the Material and methods section in subitems.
3. Table 1 should be revised, and vertical lines should be removed. Please, follow the journal’s guideline for captions of figures and tables.
4. In Figure 3, are there random dots between “Pre-Mal” and “Pos-Mal” boxplots?
5. As the treatment performed by the authors was administered only in nestling, this information should be added to the title of the manuscript.
Author Response
Why did the authors choose buzzards as experimental model? Could the authors elaborate in the introduction section?
L 54-55: We are grateful for the suggestion and added further information about the experimental model which is a part of a long-term study of Bielefeld university. See also Ll. 65-66 in materials and methods section.
For a better understanding, I highly suggest the division of the Material and methods section in subitems.
Revised as suggested.
Table 1 should be revised, and vertical lines should be removed. Please, follow the journal’s guideline for captions of figures and tables.
We are grateful for the remark and improved the style of the table.
In Figure 3, are there random dots between “Pre-Mal” and “Pos-Mal” boxplots?
We are grateful for this request. A small amount of random variation (as implemented by the geom_jitter function of R package ggplot2) was added to the location (only relative to the x-axis) of each point. This is a useful way of handling overplotting of similar values.
As the treatment performed by the authors was administered only in nestling, this information should be added to the title of the manuscript.
Revised as suggested.
Reviewer 3 Report
The authors experimentally tested the effects of Malarone (atovaquone-proguanil) on the physiology of common buzzards. Compared to control groups, they found no significant effects of treatment groups with a single dose application of Malarone on bird body condition, growth rate and chemical blood parameters. Therefore they suggest an average of 11mg/kg body weight of Malarone, as used in their study, can be tolerable by common buzzards.
Overall, I believe this study is well executed and written. The statistical methods seem appropriate to me. Their major findings may have some implications for future wild bird management. However, I do find several elements of the MS difficult to interpret in their current form.
1. The infection status of their experimental birds is unknown. Was the malarone treated to birds regardless of their infection status by single infection with Plasmodium or Leucocytozoon or Haemoproteus or by double infection? Infection status could play a critical role in avian physiology and behavior.
2. What is the biological half-life of Malarone in birds? Is an average 7-day interval long enough to examine the potential side effects gradually caused by Malarone in a longer course?
3. Figures and legends lack statistical information. Statistics between boxplots or between curves should be specified in each figure panel and major finding of each plot should be briefly summarized in the legends.
4. L35-39: confusing long sentence. Please consider rephrasing it.
5. L138-139: I did not see such results from the figures. Which plots or curve are indicators for the effects of sex or age?
6. Figure 3: what is the meaning of the dotted lines connecting two boxplots? Whether there is significant difference between boxplots should be specified in each figure panel.
Author Response
- The infection status of their experimental birds is unknown. Was the malarone treated to birds regardless of their infection status by single infection with Plasmodiumor Leucocytozoon or Haemoproteus or by double infection? Infection status could play a critical role in avian physiology and behavior.
Thanks for the request. Individuals were randomly selected, i.e. knowledge of infection status. This study primarily focused on the tolerability of Malarone, regardless of the infectious status. In discussion section we briefly included the possible bias in results in relation to blood chemistry parameters (see Ll. 272-274). In a separate study, correlation of infection status and physiological costs were investigated (see Rinaud, T., et al., Apparent physiological costs of blood parasites in the early-life of a vertebrate host-only during acute infections. 2022, https://ecoevorxiv.org/4tcqu/).
- What is the biological half-life of Malarone in birds? Is an average 7-day interval long enough to examine the potential side effects gradually caused by Malarone in a longer course?
Thanks for the request. We do not have references for the half-life of Malarone in buzzards. For humans, a short half-life of proguanil from 12 – 21h in adults and pediatric patients has been described. The elimination half-life of atovaquone is even shorter in pediatric (1 to 2 days) than in adult patients (2 to 3 days). Based on these references, we can only make comparisons to the nestlings and cannot completely rule out the possibility of changes more than 7 days after treatment. To further clarify this point, we insert a recommendation for detailed follow-up studies in the discussion section (see Ll. 225 - 228).
- Figures and legends lack statistical information. Statistics between boxplots or between curves should be specified in each figure panel and major finding of each plot should be briefly summarized in the legends.
We are grateful for the suggestion and opportunity to improve the figures and its legends and added more detailed description.
- L35-39: confusing long sentence. Please consider rephrasing it.
Revised as suggested.
- L138-139: I did not see such results from the figures. Which plots or curve are indicators for the effects of sex or age?
We thank for the opportunity to make this clearer and improved the figure and tables.
- Figure 3: what is the meaning of the dotted lines connecting two boxplots? Whether there is significant difference between boxplots should be specified in each figure panel.
We thank for the opportunity to make this clearer and added a more detailed description. Dotted lines indicate paired values obtained for the same individual but only applies for the “Pre-” and “Post-mal” condition.
Reviewer 4 Report
Review for the article: "Tolerability of atovaquone -proguanil application in common bussards".
The article is well written, the statistical inference responds to the proposed objectives. The authors were careful to treat the variables to use regression analysis. The material and methods item clarifies how the data were treated. However, for the presentation of the results, I have some suggestions below.
The line 195, Mann-Whitney tests is used to compare independente samples as was the case and not pairwise. The Wilcoxon in used for pairwise samples as was perfomed in the article and registered on line 208. Tables 2, page 6, comparts populations of distinct pre-control and pre-treatment and pos-control and pos-treatment.
Since no difference was significant in the comparison of groups table 2, I believe that just saying that there was no significant difference in the aplication of the Mann-Whitney test, and the Benjamini-Hochberg correction. The table 2 would be a suplementary table, table S2.
Table 3 presents significant values for the variables GGT and LDH and not significant for others variables. The authors can record the values in the text the article for the significant variables and cite those that were not significant. Therefore table 3 could also be a supllementary data, table S3. The adjusted value of p=0.06, in table 3, for the variables GGT and LDH can be considered significant in the authors consider it pausible. Some articles in statistics discuss and consider this possibility. I indicate the reference: Greeland, S., et al., Statistical tests, P values, confidence intervals, and power: a guide to misinterpretations> Eur. J. Epidemiol., 2015. 31:337-350.
In addition to the supllementary tables (S2 and S3), if the journal allows, the authors could attach, for readers, tables with the raw data used in the application of statistical inference analyses.
These are the observations and suggestions I have for the moment.

Author Response
Since no difference was significant in the comparison of groups table 2, I believe that just saying that there was no significant difference in the aplication of the Mann-Whitney test, and the Benjamini-Hochberg correction. The table 2 would be a suplementary table, table S2.
We are grateful for the suggestion and revised as suggested.
Table 3 presents significant values for the variables GGT and LDH and not significant for others variables. The authors can record the values in the text the article for the significant variables and cite those that were not significant. Therefore table 3 could also be a supllementary data, table S3. The adjusted value of p=0.06, in table 3, for the variables GGT and LDH can be considered significant in the authors consider it pausible. Some articles in statistics discuss and consider this possibility. I indicate the reference: Greeland, S., et al., Statistical tests, P values, confidence intervals, and power: a guide to misinterpretations> Eur. J. Epidemiol., 2015. 31:337-350.
We are thankful for this suggestion and the opportunity to explain this in greater detail. The P-value provides the probability of observing a test statistic that is as large or larger than the empirical test statistic. Hence there is a meaningful interpretation of P-values that goes beyond the discrete categorization as ‘significant’ vs ‘non-significant’ that is fully relying on the chosen alpha threshold (commonly alpha = 0.05). In the current manuscript we consistently applied a alpha threshold of 0.05 and additionally provide the true P-values for the reader.
In addition to the supllementary tables (S2 and S3), if the journal allows, the authors could attach, for readers, tables with the raw data used in the application of statistical inference analyses.
We thank for this remark. Raw data are deposited via Github (see Data Availability Statement).